

# Optical and microphysical properties of ice crystals in Arctic clouds from lidar observations

Patrick Chazette[1], Jean-Christophe Raut[2]

[1] LSCE/IPSL, CNRS-CEA-UVSQ, Paris-Saclay University, CEA Saclay, 91191 Gif sur Yvette, France

[2] LATMOS/IPSL, Sorbonne University, UVSQ, CNRS, Campus Pierre et Marie Curie, 75252 Paris, France

Correspondence to: Patrick Chazette (patrick.chazette@lsce.ipsl.fr)

**Abstract.** The vertical profiles of the optical properties, effective radius of ice crystals and ice water content (IWC) in Arctic semi-transparent stratiform clouds were assessed using quantitative ground-based lidar measurements performed from 13 to 26 May 2016 in Hammerfest (north of Norway, 70° 39′ 48″ North, 23° 41′ 00″ East). The field campaign was part of the Pollution in the ARCtic System (PARCS) project of the French Arctic Initiative. The presence of low-level semi-transparent stratus clouds was noted on 16 and 17 May, and they were sampled continuously by a ground-based Raman-$N_2$ lidar emitting at the wavelength of 355 nm. These clouds were located just above the atmospheric boundary layer where the 0°C isotherm reached around 800 m above the mean sea level (a.m.s.l.). To ensure the best penetration of the laser beam into the cloud, we selected case studies with cloud optical thickness (COT). Lidar-derived multiple scattering coefficients were found to be close to 1 and ice crystal depolarization around 10%, suggesting that ice crystals were small and had a rather spherical shape. This agrees with our Mie computations determining effective radii between ~5 and 20 µm in the clouds for ice water contents between 1 and 8 mg m$^{-3}$, respectively. Direct estimate of the microphysical parameters of ice clouds via lidar measurements is a significant asset for the study of their large-scale radiative impact, while reducing the need for experimental resources.

Keywords: Arctic stratiform clouds, lidar, optical properties, ice microphysical properties, effective radius, ice water content

## 1 Introduction

Cloud radiative effects significantly influence the radiative budget of the planet at high latitudes (Kay and Gettelman, 2009; Schweiger et al., 2008; Morrison et al., 2012), particularly beyond the Arctic Circle, and contribute to the well-known "Arctic amplification" phenomenon (Serreze and Barry, 2011; Liu and Key, 2014; Wendisch et al., 2019). In particular, clouds have


impacted sea ice and glaciers which have been melting for several years in connection with climate change (Serreze et al., 2009; Hansen et al., 2010; Koenigk et al., 2013).

Arctic clouds exhibit a robust annual cycle with maximum cloudiness in fall and minimum cloudiness in winter (Yu et al., 2019). The annual cloud fraction amounts to about 80%, with

predominant low-level clouds up to 70% of the time from spring to autumn (Curry and Ebert, 1992). Based on observations from spaceborne LITE (Lidar In-space Technology Experiment), GLAS (Geoscience Laser Altimeter System) and CALIOP (Cloud-Aerosol LIdar with Orthogonal Polarization) lidar, Berthier et al. (2008) showed that those low-level stratiform clouds represent between 30 and 40% of the cloud cover. As they are located in air masses

below the 0°C isotherm, these clouds are very often composed of ice crystals. The presence of supercooled liquid water droplets at the top of these clouds has nevertheless often been reported (Morrison et al., 2012; Shupe et al., 2008). Their subgrid-scale treatment is usually underestimated by physical parameterizations used in the climate models (Klaus et al., 2016; Taylor et al., 2019). This leads to a wrong representation of surface radiative fluxes, both in the

solar and infrared spectra (Harrington et al., 1999; Klein et al., 2009; Maillard et al., 2021; Koenigk et al., 2013; Di Biagio et al., 2021). Reliable observations of microphysical and optical properties of ice particles, in particular of their ice crystal effective sizes, are critical for the evaluation of cloud parameterizations and for determining how clouds impact radiation (Zhang et al., 1996). Harrington and Olsson (2001) showed changes of up to 80 W m$^{-2}$ due to a variation

in the effective radius of ice particles. Accurately representing ice clouds in models is therefore necessary to realistically simulate the evolution of the Arctic surface energy budget.

The accuracy of cloud observations has been greatly improved in recent decades by the widespread use of lidar observations so as to better understand and predict the Earth system climate (e.g. Hoffmann et al., 2009). Since the precursory work of Platt (1977), ground-based

lidar observations have made it possible to better characterize the vertical distribution of cloud structures (Sassen, 1991). Coupled with infrared radiometric observations and even radar measurements, microphysical properties, such as effective radius of ice crystals, are now accessible (Kalesse et al., 2016). With aircraft and satellite-based instrumentation, microphysical properties are now rendered at a larger spatial scale (Delanoë and Hogan, 2010;

Chazette et al., 2022; Liu et al., 2012; Lampert et al., 2009). However, the velocity of those instrumented platforms, along with their temporal resolution, prevent from sampling clouds at a spatial resolution which would be adequate to properly highlight their internal structure. Indeed, the resulting data provide cloud properties typically averaged over 1 km, which may be



insufficient to study cloud processes at a microphysical scale. Examining cloud structures with a high vertical resolution is however possible via ground-based lidars. Such instruments are unable to match the level of detail from aircraft in situ measurements or the spatial coverage of satellites, but they are fittingly positioned to capture the diurnal variability with high temporal and vertical resolutions (Shupe et al., 2011). This remains a challenge, as it requires that lidar measurements penetrate through the cloud with low attenuation. Semi-transparent clouds with small optical thicknesses ($\lesssim 2$) are therefore accessible to lidar measurements.

Due to the growing interest of the climate science community in clouds over the Arctic region, numerous experiments have been set up recently. For instance the ACLOUD (Arctic Cloud Observations Using airborne measurements during polar Day (Ehrlich et al., 2019)) in 2017, AFLUX (Airborne measurements of radiative and turbulent FLUXes of energy and momentum in the Arctic boundary layer (Mech et al., 2022)), PASCAL (Physical feedback of Arctic PBL, Sea ice, Cloud And AerosoL (Wendisch et al., 2019)), both in 2019 or MOSAiC (The Multidisciplinary Drifting Observatory for the Study of Arctic Climate (Shupe et al., 2022)) in 2019-2020. In this respect, we conducted Raman lidar observations at 355 nm during the PARCS (Pollution in the ARCtic System) field experiment (Chazette et al., 2018; Totems et al., 2019) in May 2016 at Hammerfest (Norway, over 70° N) in order to better constrain the estimate of ice crystal properties in the vertical structure of stratiform clouds. The high vertical resolution of the lidar allowed to highlight the presence of supercooled water pockets embedded into ice clouds and to trace the vertical profiles of the optical properties of semi-transparent stratiform clouds. It is then possible to derive the hydrometeor size profiles taking into account the multiple scattering effects.

The lidar system and associated theory are presented in Section 2. Section 3 describes the methodology used to retrieve the effective radii of ice crystals and ice water content from the lidar measurements. The optical properties of the ice crystals accessible to the Raman lidar measurements are given in Section 4 and their associated effective radius in Section 5. Finally, the conclusions summarizing the findings are provided in Section 6.

## 2    Ground-based lidar observations

### 2.1    Instrument

The ground-based WALI (Water vapor and Aerosol Lidar) was the Raman lidar operated during the PARCS campaign (Totems et al., 2019; Chazette et al., 2018). It uses an emitted wavelength of 354.7 nm and is designed to fulfil eye-safety conditions. The instrument calibration and the associated errors are documented in Chazette et al. (2014), and more recently in Totems et al.



(2021). During the field experiment, the acquisition was performed continuously with a vertical resolution of 15 m for mean profiles of 1000 laser shots, leading to a temporal sampling close to 1 min. The overlap function of this lidar is ~200 m and can be assessed experimentally for lidar signal correction in the lowest atmospheric layers.

Three different channels have been used to study ice-clouds: the first one to detect the total (co-polarized and cross-polarized with respect to the polarized laser emission) backscatter coefficients, the second to detect solely the cross-polarized backscatter coefficients, and the third to detect the vibrational Raman scattering on nitrogen. The field of view of the lidar is less than 2 mrad to minimize the effect of multiple scattering within the troposphere.

In the following, the lidar equation is applied to the specific case of stratiform clouds.

### 2.2   Approach

Using Raman lidar sampling of stratiform ice clouds, we aim to retrieve the vertical profiles of the optical and structural properties of ice crystals. This type of data remains very sporadic and is sorely lacking for a realistic modelling of the climatic impact of ice clouds.

Our approach is the following: i) isolate the cloud structures that are entirely sampled by the lidar laser beam, ii) assess the cloud optical thickness (COT) via the elastic and $N_2$-Raman channels and check the consistency of the result, iii) compute the integrated apparent backscatter coefficient of the cloud to derive the effective lidar ratio (LR), product of the LR by the multiple scattering coefficient, iv) compute the vertical profiles of extinction, backscatter

coefficients and ice crystal depolarization ratio (ICDR) in the cloud, and v) invert Mie calculations to estimate the effective size of the ice crystal and ice water content of the selected cloud layers.

### 2.3   Theory

After the molecular transmission is corrected, the elastic lidar signal in the form of total

apparent backscatter coefficient ($\beta_{app}$) at zenith viewing beyond the influence of the overlap factor (~200 m) is written at the altitude $z$ in the cloud as:

$$\beta_{app}(z) = C \cdot \underbrace{\left(\beta_m(z) + \beta_a(z) + \beta_c(z)\right)}_{\beta(z)}$$

$$\cdot \exp\left(-2 \cdot \eta(z) \cdot \underbrace{\int_{z_b}^{z} \alpha_c(z') \cdot dz'}_{COT(z_b,z)}\right) \cdot \exp\left(-2 \cdot \underbrace{\int_{z_0}^{z_b} \alpha_a(z') \cdot dz'}_{AOT(z_0,z_b)}\right) \tag{1}$$





where $C$ is the lidar system constant, $\beta_m$ and $\beta_a$ are the backscatter coefficients due to molecules and aerosols, respectively. The cloud backscatter coefficient is represented by $\beta_c$. The sum of the backscatter coefficients is the total volume backscatter coefficient ($\beta$). The vertical profile of the COT is calculated by integrating the extinction coefficient of ice crystal ($\alpha_c$) between the

cloud base ($z_b$) and altitude $z$ in the cloud. Equation (1) assumes that the extinction coefficient due to aerosols is negligible above $z_b$, which is realistic during the PARCS campaign (Chazette et al., 2018). The aerosol optical thickness (AOT) is therefore calculated under the cloud base from lidar altitude $z_o$ to $z_b$ as the integral of the aerosol extinction coefficient ($\alpha_a$). The multiple scattering coefficient $\eta$ as defined by Platt (1981) traces the optical path lengthening associated

with the multiple scattering process by the hydrometeors. The system constant $C$ is obtained in cloud-free condition via the synergy between the lidar elastic and Raman channels above the planetary boundary layer (PBL) top, where only molecular scattering occurs, with a resultant relative accuracy of 5%.

For the N$_2$-Raman channel, after correction for molecular transmission and normalization to

atmospheric density, the lidar signal $\beta_{appN2}$ is written in a proportionality relationship as

$$\beta_{appN2}(z) \propto exp\left(-\eta(z) \cdot \left(1 + \left(\frac{387}{355}\right)^{-A_c}\right) \cdot \underbrace{\int_{z_b}^{z} \alpha_c(z') \cdot dz'}_{COT(z_b,z)}\right)$$

$$\cdot exp\left(-\left(1 + \left(\frac{387}{355}\right)^{-A_a}\right) \cdot \underbrace{\int_{z_0}^{z_b} \alpha_a(z') \cdot dz'}_{AOT(z_0,z)}\right) \qquad (2)$$

where $A_c$ ($A_a$) is the Ångström exponent in the cloud (aerosol layer) between the wavelengths of 355 and 387 nm. Both Ångström exponents are supposed to be constant. It is reasonable to assume $A_c$ to be close to 0. It is also assumed here that the multiple scattering coefficient in clouds is the same at 355 and 387 nm.

**2.3.1    Cloud optical thickness**

Using the elastic scattering channel, the ratio of $\beta_{app}$ above the top ($z_t$) and below the base ($z_b$) of the cloud, where molecular diffusion is assumed to be the only one contribution to the lidar signal (negligible AOT), can be written as:



$$\frac{\beta_{app}(z_t)}{\beta_{app}(z_b)} = \frac{\beta(z_t)}{\beta(z_b)} \cdot exp\left(-2 \cdot \eta(z_t) \cdot \underbrace{\int_{z_b}^{z_t} \alpha_c(z') \cdot dz'}_{COT(z_b,z_t)}\right) \tag{3}$$

Hence, we can assess the COT between $z_b$ and $z_t$ by

$$COT(z_b, z_t) = \frac{1}{2 \cdot \eta(z_t)} \cdot \log\left(\frac{\beta(z_t) \cdot \beta_{app}(z_b)}{\beta(z_b) \cdot \beta_{app}(z_t)}\right) \tag{4}$$

The COT can also be determined directly from the $N_2$-Raman channel by a ratio at the same altitudes. This leads to the relationship:

$$COT(z_b, z_t) \approx \frac{1}{\left(1 + \left(\frac{387}{355}\right)^{-A}\right) \cdot \eta(z_t)} \cdot \log\left(\frac{\beta(z_t) \cdot \beta_{app}(z_b)}{\beta(z_b) \cdot \beta_{app}(z_t)}\right) \tag{5}$$

Accurate determination of the COT requires knowing the multiple scattering coefficient $\eta(z_t)$. At this stage, we do not know this value, so we can only calculate the effective COT ($COT_e = \eta \cdot COT$) instead, which includes multiple scattering processes. Furthermore, it should be noted immediately that if independent determinations of the $COT_e$ by the elastic and Raman-$N_2$ channels match, this means that the assumption of free-aerosol altitude zones below and above
the cloud is justified. Indeed, in this case, aerosols have little influence on the determination of $COT_e$ via the $N_2$-Raman channel (Section 2.3.2).

### 2.3.2  Vertical profiles of extinction and lidar ratio

The simultaneous use of the $N_2$-Raman and elastic channels makes it possible to find the vertical profiles of the extinction and backscatter coefficients associated with the clouds. Their LR is a
characteristic parameter of the scattering layers (Chazette et al., 2016).

By normalizing to the altitude at the cloud base $z_b$ for each altitude z, equation (2) leads to the relationship:

$$\exp(2 \cdot \eta(z) \cdot COT(z_b, z)) \approx \frac{\beta_{appN2}(z_b)}{\beta_{appN2}(z)} \tag{6}$$

Combining with equation (1), the cloud backscatter coefficient between $z_b$ and $z_t$ can be derived as:



$$\beta_c(z) \approx \frac{\beta_{app}(z)}{C} \cdot \frac{\beta_{appN2}(z_b)}{\beta_{appN2}(z)} \cdot \exp(2 \cdot AOT(z_o, z_b)) - \beta_m(z) \tag{7}$$

AOT is known, as it is assessed outside the cloud. Chazette et al. (2018) have shown that its value is small, lower than 0.04 at 355 nm in the PBL.

From (7) we derive also $\forall z \in [z_b, z_t]$:

$$\underbrace{\eta(z) \cdot COT(z_b, z)}_{COT_e(z_b, z)} \approx \frac{1}{2} \cdot log\left(\frac{\beta_{appN2}(z_b)}{\beta_{appN2}(z)}\right) \tag{8}$$

$COT_e(z_b, z)$ is the effective cumulative optical thickness profile that tends towards the $COT_e$ of

the cloud when z is close to $z_t$. Its derivative with respect to $z$ gives the effective extinction coefficient $\alpha_c^e(z)$ from which we obtain the vertical profile of the effective LR ($LR_e$), which is the ratio of $\alpha_c^e$ to $\beta_c$:

$$\begin{cases} \alpha_c^e(z) = \dfrac{\partial COT_e(z_b, z)}{\partial z} \\ \underbrace{\eta \cdot LR(z)}_{LR_e(z)} = \dfrac{\alpha_c^e(z)}{\beta_c(z)} \end{cases} \tag{9}$$

It is worth noting that given the degrees of freedom of the equation system, it is only the effective values that are assessed.

### 2.3.3  Integrated backscatter properties

The effective LR ratio equivalent to the scattering layer ($\widetilde{LR}_e$) can be directly evaluated by using the integration of $\beta_{app}$ ($\beta_{int}$) over the height of the cloud. This can be performed by comparing the value deduced from the observations with the one retrieved from the theoretical expression.

The theoretical expression of the integrated apparent backscatter coefficient $\beta_{int}$ is calculated

at altitude $z$ from:

$$\beta_{int}(z_b, z) = \int_{z_b}^{z} \beta_{app}(z') \cdot dz' \tag{10}$$

Assuming $\beta_m(z) \ll \beta_c(z)$, we have:

$$\beta_{int}(z_b, z) \approx \int_{z_b}^{z} C \cdot \beta_c(z') \cdot \exp\left(-2 \cdot \eta(z') \cdot \int_{z_b}^{z'} \alpha_c(z'') \cdot dz''\right) \cdot dz' \tag{11}$$

Assuming constant values for LR and $\eta$ between $z_b$ and $z$, an integration by parts leads to the relationship:





$$\beta_{int}(z_b, z) \approx \frac{C}{2 \cdot \eta \cdot \widetilde{LR}_e} \cdot [1 - \exp(-2 \cdot \eta * COT(z_b, z))] \tag{12}$$

Another method is to compute $\widetilde{LR}_e$ by weighting $LR_e$ by the cloud backscatter coefficient:

$$\widetilde{LR}_e = \frac{\int_{z_b}^{z_t} \beta_c(z) \cdot LR_e(z) \cdot dz}{\int_{z_b}^{z_t} \beta_c(z) \cdot dz} \tag{13}$$

### 2.3.4    Ice crystal linear depolarization ratio

The linear volume depolarization ratio ($VDR$) is calculated via the ratio of perpendicularly and parallelly polarized channels defined in respect to the laser emission as in Chazette et al. (2012a) where the different sources of uncertainty are discussed. The ICDR is calculated using a similar relationship to that used for aerosols (Chazette et al., 2012), from $VDR$ and $\beta_c$:

$$ICDR(z) = \frac{\beta_m(r) \cdot (VDR_m - VDR(z)) - \beta_c(z) \cdot VDR(z) \cdot (1 + VDR_m)}{\beta_m(r) \cdot (VDR(z) - VDR_m) - \beta_c(z) \cdot (1 + VDR_m)} \tag{14}$$

where the molecular linear volume depolarization ratio ($VDR_m$) has been taken equal to 0.3945% at 355 nm following Collis and Russel (1976) for Cabannes scattering.

### 2.3.5    Multiple scattering coefficient

To evaluate $\eta$, we used a Monte Carlo model specially developed for lidar measurements. This model was used to analyse the LITE (Lidar In-Space Technology Experiment) measurements by Berthier et al. (2006). It has also been used to estimate multiple scattering through forest canopy (Shang and Chazette, 2015) for airborne and spaceborne lidar measurements. The outputs of the model were successfully compared with the results of Wiegner et al. (1997) and again compared to simulations performed using the photon variance-covariance (PVC) method for quasi-small-angle multiple scattering (Hogan, 2006). The PVC algorithm can represent anisotropic phase function in the near 180° direction. As we noticed a very small difference in $\eta$ (< 5%) between the two modelling approaches, we can infer both approaches mutually validate. The Monte Carlo model was initialized with the ice crystal phase functions determined by the method presented below, themselves constrained by the lidar measurements.

## 3    Method for the determination of ice water content and ice crystal effective radius

In order to assess the vertical profiles of ice crystal effective radius ($r_{eff}$) and ice water content (IWC), we use a Mie code assuming ice particles are all spherical, as in many two-moment microphysical schemes (Milbrandt and Yau, 2005; Morrison and Gettelman, 2008; Thompson et al., 2008). The complex index of refraction of ice is taken from the database established by

Warren and Brandt (2008) from wavelengths between 44 nm and 2 µm at 266 K. The refractive index of ice crystals has been interpolated against the logarithm of the wavelength and assessed to be equal to 1.324 at the wavelength of the lidar (355 nm), with a negligible imaginary part.

Cloud ice crystal size distributions $\frac{dN}{dr}$ are commonly represented by generalized Gamma distribution (Stephens et al., 1990), as in microphysical schemes most widely used (Morrison et al., 2005; Thompson et al., 2008) :

$$\frac{dN}{dr} = 2^{(1+\mu_i)} \cdot N_{0i} \cdot r^{\mu_i} \cdot e^{-2 \cdot \lambda_i \cdot r} \tag{15}$$

where $r$ is the radius of the ice crystal and d$N$ the number concentration of ice crystals between $r$ and $r$+d$r$. The intercept, slope and shape parameters of the size distribution are $N_{0i}$, $\lambda_i$ and $\mu_i$, respectively. The normalised size distribution $n(r)$ of the ice crystal can also be written equivalently as a function of $r$ and the effective radius $r_{eff}$ (Hansen and Travis, 1974):

$$n(r) = \frac{1}{N_t}\frac{dN}{dr} = \frac{1}{(v_{eff} \cdot r_{eff})^{\frac{1}{v_{eff}}-2}} \cdot \Gamma\left(\frac{1}{v_{eff}} - 2\right) \cdot r^{\frac{1}{v_{eff}}-3} \cdot e^{-\frac{r}{v_{eff} \cdot r_{eff}}} \tag{16}$$

where $v_{eff}$ is the effective variance of the distribution and $\Gamma$ the Euler-Gamma function. The total number of ice crystals is symbolized by $N_t$.

Here, ice crystals are assumed to be spheres. In this paper, to evaluate the sensitivity of the effective variance $v_{eff}$ of ice crystals size distribution on cloud optical properties, we test two different values. First, the shape parameter $\mu_i$ is assumed to be zero (Marshall-Palmer distribution), as in many well-used two-moment microphysical schemes (Morrison et al., 2005; Thompson et al., 2008). The effective variance $v_{eff}$ is therefore $v_{eff} = \frac{1}{\mu_i+1} = \frac{1}{3}$. Second, we consider a smaller value of the effective variance ($v_{eff} = 0.2$), corresponding to a larger shape parameter ($\mu_i = \frac{1}{v_{eff}} - 3 = 2$), which reduces the backscattering, then the LR, for the smallest particles (Fig. 1).

**Effective radius.** The extinction and backscattering cross-sections are calculated using a Mie code as a function of the ice crystal radius in the range 1 to 200 µm. For a series of $r_{eff}$ values ranging from 5 to 40 µm, the extinction and backscatter coefficients are then derived by integrating their respective cross sections over the size distribution $n(r)$, enabling the retrieval of LR for each value of $r_{eff}$. By definition, LR is independent of the assumed value of the total number of ice crystals $N_t$. Figure 1 shows the variation of LR as the function of $r_{eff}$ following





Mie calculation for $\nu_{eff}$ equal to 0.20 and 0.33. LR turns out to be a monotonically decreasing function of $r_{eff}$. By minimising the discrepancies between the theoretical and measured values of LR, the effective radius of ice crystals can then unequivocally be determined for the considered size distribution range.

5    The parameters of the size distribution are thus derived as:

$$\begin{cases} \lambda_i = \dfrac{1}{2\,\nu_{eff}\cdot r_{eff}} \\ N_{0i} = \dfrac{N_{ti}\cdot\lambda_i{}^{\mu_i+1}}{\Gamma(\mu_i+1)} \end{cases} \tag{17}$$

where $N_t$ is the total number of ice crystals obtained such that the theoretical extinction coefficient matches its observed counterpart.

**Ice water content.** Finally, the IWC is obtained using:

$$IWC = \frac{\pi}{6}\cdot\rho_i\cdot N_{0i}\frac{\Gamma(\mu_i+4)}{\lambda_i{}^{\mu_i+4}} \tag{18}$$

10   where $\rho_i$ = 920 kg m$^{-3}$ is the density of pure ice.

The procedure is repeated at each altitude sampled, enabling us to derive vertical profiles of $r_{eff}$ and IWC for each case study.

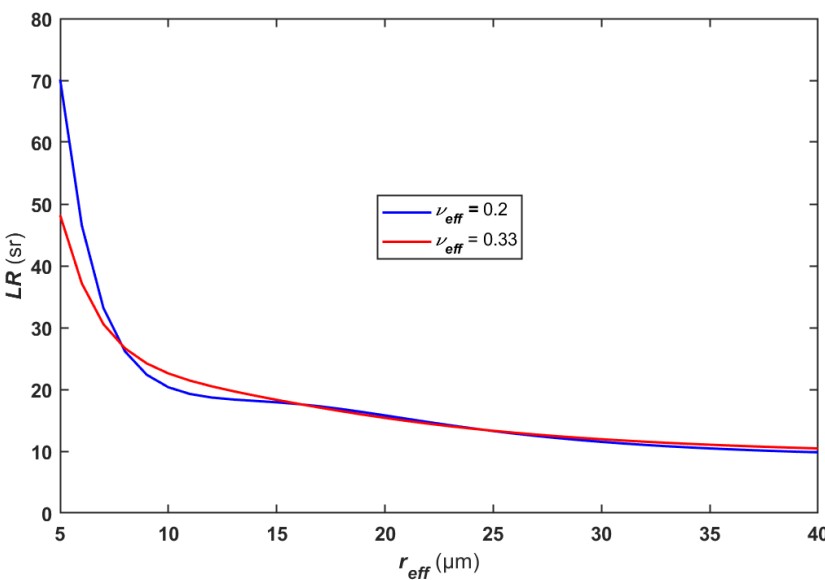


Figure 1. Variation of the theoretical lidar ratio (*LR*) obtained from Mie calculations as a function of the effective radius ($r_{eff}$) of ice crystals for two values of the effective variance $v_{eff}$.

## 4 Lidar sampling

### 4.1 Description

Just before the arrival of the stratiform clouds over the measurement site, ultralight aircraft flights were made with a payload including: i) a meteorological probe to measure temperature, relative humidity, and pressure, and ii) the Lidar for Automatic Atmospheric Survey Using Raman Scattering (LAASURS) for the retrieval of the aerosol extinction coefficient. This payload is described in Chazette et al. (2018). On 16 May 2016, 11:30 local time (LT), the vertical profiles of temperature, relative humidity and aerosol extinction coefficient derived from the ultralight payload are given in Fig. 2. That latter is negligible over 1.2 km above mean sea level (a.m.s.l.) (Chazette et al., 2018). The 0°C isotherm is reached at an altitude of 0.8 km a.m.s.l. with relative humidity increasing with altitude. With the advection of moist air masses over the site, the formation of ice clouds is therefore highly probable above 0.8 km a.m.s.l. This value is within the range of the bottom height of boundary layer mixed phase clouds in the western Arctic (McFarquhar et al., 2007; Maillard et al., 2021).

In the early afternoon of 16 May, the sky became overcast, leading to thick stratiform clouds around 16:00 LT. These clouds were sampled between 16 and 17 May by the ground-based lidar. The apparent backscatter ratio ($\beta_{app}/\beta_m$) and VDR time series are given in Fig 3. The stratiform clouds extend between ~0.8 and 6 km a.m.s.l. and display complex structures highlighting their great heterogeneity. This heterogeneity is certainly linked to the formation processes of the hydrometeors as well as to the variability of the atmospheric dynamics in connection with wind shear. The VDR is also highly variable temporally and vertically. The higher values (>8%) indicate the location of frozen hydrometeors, while the lower values inside clouds (< 1%) may indicate the presence of supercooled water liquid droplets or molecular holes in the cloud structure. It is worth noting that, as the cloud-related scattering coefficient is very large compared to that of the molecules and aerosols, the VDR should not be very far from the PDR. Our observations of liquid structures embedded in ice clouds are in agreement with previous field measurements (Rangno and Hobbs, 2001; Korolev and Isaac, 2003) which suggested that different pockets of solely water or ice in mixed-phase regions coexist with typical scale of tens of meters. In contrast, large-scale models erroneously assume that liquid and ice phases are uniformly mixed within each model grid box (Tan and Storelvmo, 2016),



with implications on the efficiency of the Wegener-Bergeron-Findeisen effect (Beesley and Moritz, 1999; Tan and Storelvmo, 2019).

The clouds are not formed over the site but over the ocean. They are then transported with a vertical gradient of the horizontal wind that generates some of the noticeable structures, such as the comma-like configuration between 00:00 and 04:00 on 17 May (Fig. 3). Between 05:00 and 07:00 LT, a cloud layer composed of supercooled droplets can be observed between 2 and 2.5 km a.m.s.l. In this case, $\beta_{app}$ is higher than 10 with a very low VDR. We also note the probable presence of supercooled liquid droplets on 16 May between 18:30 and 19:30 LT around 2.5 km a.m.s.l. Estimating the properties of ice crystals therefore requires selecting profiles outside supercooled water pockets.

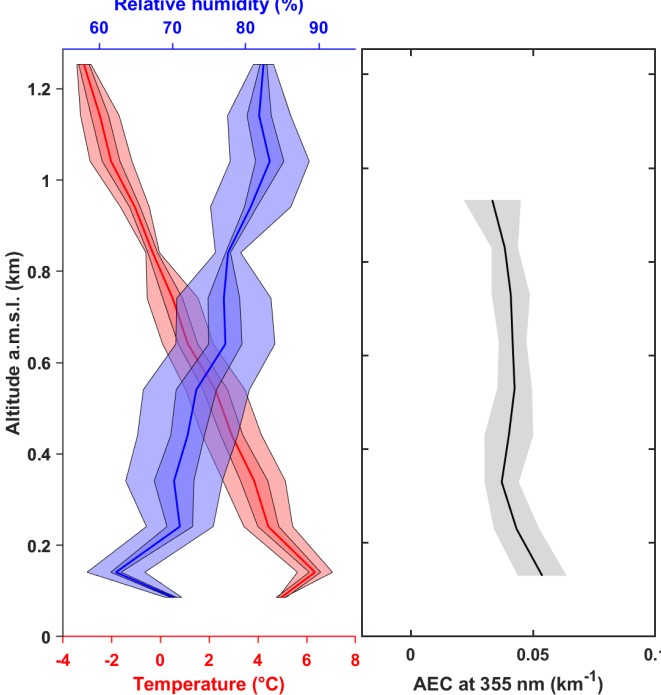

Figure 2. Vertical profiles of temperature, relative humidity, and aerosol extinction coefficient (AEC) derived from the ultralight on 16 May 2016, 11:30 LT. The light shaded areas give the data variability in 100 m-thick atmospheric layers. The darker colored areas give the error on temperature (red) and relative humidity (blue). The grey shaded area is the error on the AEC.



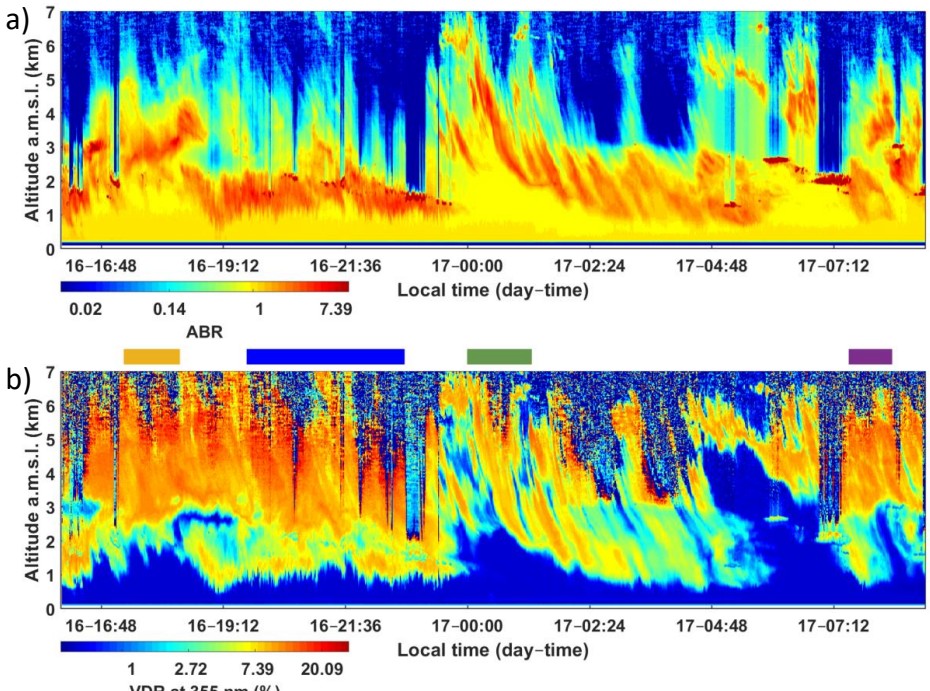

Figure 3. Time series of the a) apparent backscatter ratio (ABR) and b) linear volume depolarization ratio (VDR) derived from the ground-based lidar. Cases studied specifically are highlighted by the color bar above each period concerned: case 1 in orange, case 2 in blue, case 3 in green and case 4 in violet.

## 4.2 Meteorological synoptic conditions

From 14 to 16 May, two ridges block a low-pressure system between the Barents and Norwegian Seas. This weather situation is illustrated by the geopotential height at 850 hPa (~1.5 km a.m.s.l.) in Fig. 4a, where the wind field is superimposed. The first ridge is located over Greenland and extends from the mid-latitudes to the Pole, while the second ridge is located between 55 and 90° E and passes over the Novaya Zemlya archipelago. The latter weakened from 16 to 17 May and thus favored the development of a low-pressure system around the Svalbard archipelago. This low then spreads to the south of the Norwegian Sea. In the afternoon of 16 May, Hammerfest was on the edge of the low (Fig. 4b) with stratified cloud structures, before it spreads to cover the town completely by the late morning of 17 May (Fig. 4c). This leads to a significant evolution of the cloud cover towards denser and precipitating clouds which appear on the afternoon of 17 May over Hammerfest. The retrieval of the optical parameters of ice crystal relies on the lidar observations of semi-transparent stratiform frozen clouds before





the arrival of denser precipitating clouds. The meteorological situation corresponds to a warm south-western regime as named by Mioche et al. (2017).

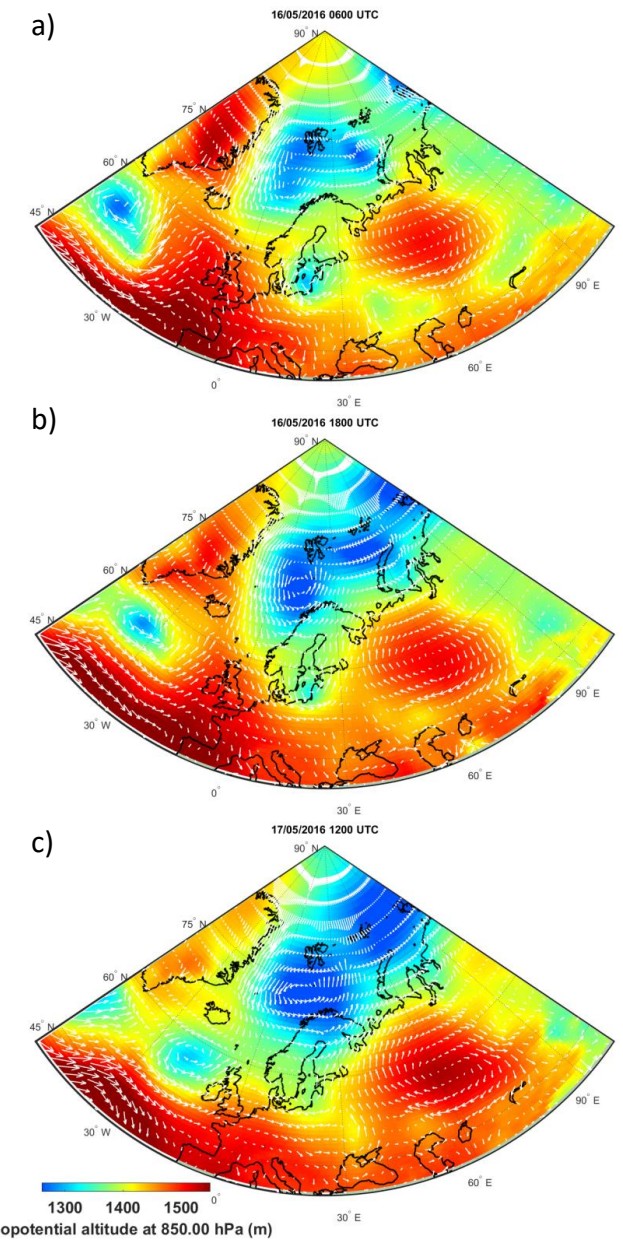

Figure 4. Geopotential height at 850 hPa on a) 16 May 2016, 06:00 UTC, b) 16 May 2016, 18:00 UTC, and 17 May 2016, 12:00 UTC. The wind field is also represented by white arrows.

### 4.3 Lidar derived optical properties

We selected 4 cases (Fig. 3) when the laser beam passed through stratiform ice clouds. For each selected period we averaged the profiles over time intervals shown in Fig. 3 to increase the signal to noise ratio of the lidar measurements. The lidar-derived integrated optical parameters of sampled clouds are given in Table 1 for each case. The optical properties are assuming a

contribution from multiple scattering which is evaluated later. The consistency of the retrievals can be assessed by comparing $COT_e$ derived from both elastic and $N_2$-Raman channels, which are two independent measurements. This ensures that the assumptions for molecular scattering before and after the cloud are reasonable. In Table 1, the "Klett" columns give the result of the Klett (1981) inversion using $\widetilde{LR}_e$ calculated via the lidar elastic (equation 12) and $N_2$-Raman

(equation 13) channels, respectively. The differences observed with the Klett inversion are directly attributable to the noise level on the upper part of the clouds, mainly for the $N_2$-Raman channel. They remain nevertheless lower than 30% between fundamentally different approaches. The stable Klett inversion significantly reduces the errors as one moves away from the top of the cloud layer, but it requires an a priori assumption on the LR.

The value of $\widetilde{LR}_e$ derived from the integral of the apparent backscatter coefficient ($\beta_{int}$, Equation 12) is shown to be very stable from one case to another (between 14 and 17 sr). It is lower than that determined from the $N_2$-Raman channel (between 21 and 25 sr, Equation 13). This difference may have several origins: i) the level of signal noise above the cloud layer that affects the boundary condition and ii) the smaller range of the $N_2$-Raman channel that therefore

integrates less of the upper part of the clouds. Nevertheless, the observed deviations on LR remain within what is generally assessed (uncertainty lower than 10 sr).

In the following, we therefore use the coupling between the elastic and $N_2$-Raman channels to determine the vertical profiles of LR and then the vertical profiles of extinction coefficient of ice crystals. For the 4 cases, the vertical profiles derived from the lidar measurements are shown

in Fig. 5. $\widetilde{LR}_e$ are mainly between 10 and 30 sr for the cloudy structures. The higher values observed in the lowest layers are related to the transition with the atmospheric boundary layer where there can be a contribution of aerosols when the cloud structures have a very low density. ICDR is retrieved between 5 and 15% in the clouds, mostly below 10%, as shown in Fig. 6 for the 4 cases. Such values are weak for large non-spherical ice crystals. We can therefore infer

that the stratus clouds observed are composed of small ice crystals with a rather spherical shape.

Assuming that the retrieved optical properties are slightly influenced by multiple scattering and considering the processes presented in Section 3, we can constrain the size distribution of the ice crystals. The phase functions calculated using Mie model, coupled to the extinction profile

$\alpha_c^e$ then allow us to evaluate the importance of multiple scattering using a Monte Carlo model. The retrieved values of $\eta$ remain above 0.95 for all cases and mainly influence the top part of the clouds. We can therefore make the reasonable assumption that multiple scattering has little influence on our results. Note that the derived values are close to those previously determined

5     from ground-based lidar measurements for the same type of clouds in the Arctic region ($\eta$ = 0.92±0.03) by Mariage et al. (2017).

Table 1. Equivalent integrated optical parameters at 355 nm for 4 case studies of cloud layers: $COT_e$ from the elastic and $N_2$-Raman channels (altitude range given in brackets), and equivalent effective LR ($\widetilde{LR_e}$). The "Klett" column shows results using the Klett (1981) inversion. Altitude

10     ranges where calculations are made are given in brackets.

| Case | Time range (LT) | $COT_e$ Elastic channel | | | $COT_e$ $N_2$-Raman channel | | |
|---|---|---|---|---|---|---|---|
| | | Equation 12 | Klett | $\widetilde{LR_e}$ (sr) | Equation 13 | Klett | $\widetilde{LR_e}$ (sr) |
| 1 | 16 May 2016 17:15 - 18:20 | 1.84 (1.3 - 6.3 km) | 1.69 | 17 | 1.79 (1.3 - 4.8 km) | 1.38 | 25 |
| 2 | 16 May 2016 19:40 - 22:45 | 2.08 (0.7 - 5.9 km) | 1.96 | 17 | 2.28 (0.7 - 5.26 km) | 2.00 | 21 |
| 3 | 17 May 2016 00:00 - 01:15 | 1.12 (2 - 6.9 km) | 0.90 | 17 | 1.15 (2 - 6.0 km) | 0.91 | 25 |
| 4 | 17 May 2016 07:30 - 08:20 | 1.51 (0.55 - 6.0 km) | 1.36 | 14 | 1.28 (0.55 - 4.5 km) | 1.01 | 21 |

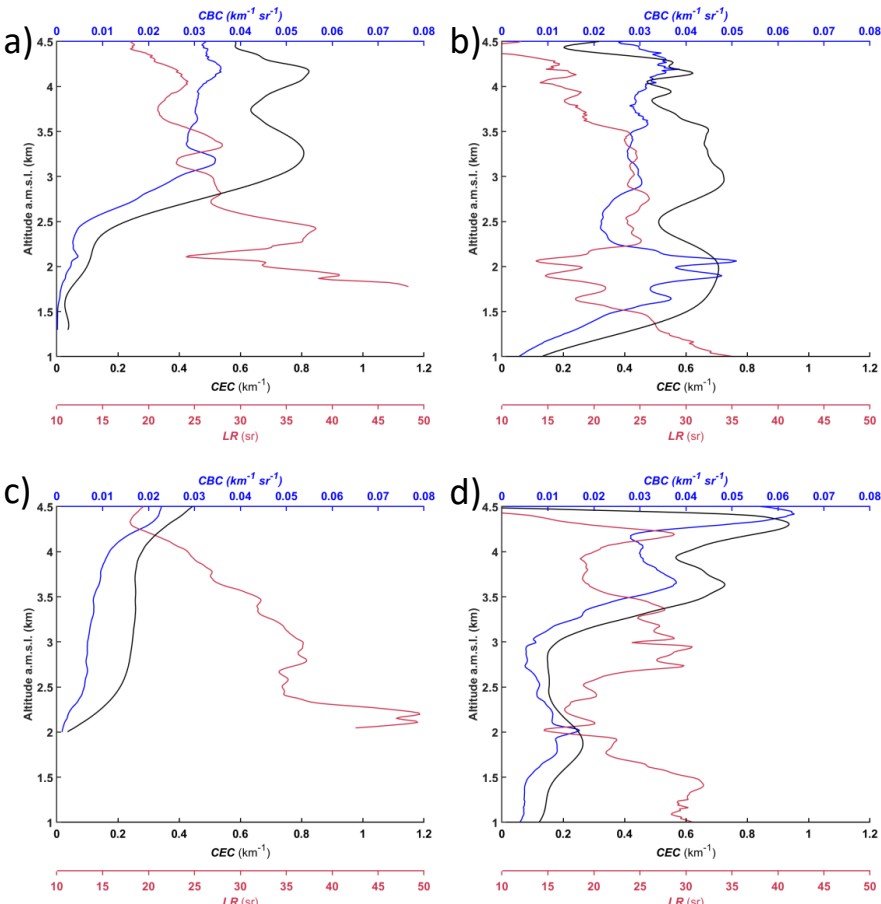

Figure 5. Vertical profiles of lidar-derived cloud extinction coefficient (CEC), lidar ratio (LR) and cloud backscatter coefficient (CBC) on a) 16 May 2016, 17:15 - 18:20 local time (LT), b) 16 May 2016 19:40 - 22:45 LT, c) 17 May 2016 00:00 - 01:15 LT, and d) 17 May 2016 07:30 - 08:20 LT.

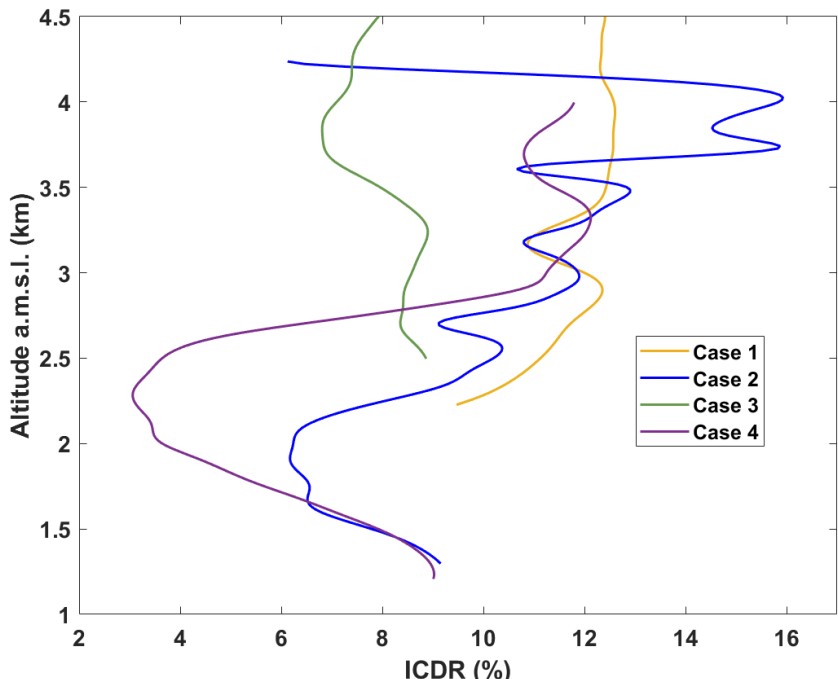

Figure 6. Vertical profiles of lidar-derived linear ice crystal depolarization ratio (ICDR) on a) 16 May 2016, 17:15 – 18:20 local time (LT), b) 16 May 2016 19:40 – 22:45 LT, c) 17 May 2016 00:00 – 01:15 LT, and d) 17 May 2016 07:30 – 08:20 LT.

## 5    Vertical profile of cloud microphysical properties

The vertical profiles of the effective radius $r_{eff}$ and ice water content calculated as in Section 3 are shown in Fig. 7. Two values of $\nu_{eff}$ are considered (0.33 and 0.20) to evaluate potential ranges of $r_{eff}$ and $IWC$. The effective radius $r_{eff}$ presents rather small values, between 5 and 20 µm in the first 4.5 km. The corresponding $IWC$ is lower than 8 mg m⁻³, as it corresponds to small ice crystals and semi-transparent stratus clouds. It can be noted that the value of $\nu_{eff}$ has little influence on $r_{eff}$ and $IWC$ because the backscatter phase functions do not change significantly (Fig. 1).

The values of $r_{eff}$ are in the lowest range of those reported in the literature for the Arctic region by  Mioche et al. (2017) who have analyzed the vertical distribution of microphysical properties of low-level single-layer mixed-phase clouds using in situ measurements from four airborne spring campaigns in the European Arctic between 2004 and 2010. For the south-western regime, they showed that the ice phase dominates the microphysical properties with mean

values of ~25 µm and less than 25 mg m$^{-3}$ for effective radius and IWC, respectively. Their analysis also revealed that large values of the liquid water content and high concentrations of small droplets may be linked to polluted situations and air mass origins from the south, leading to the lower values of ice crystal size and $IWC \sim 10$ mg m$^{-3}$. Our results are therefore consistent

with airborne measurements on the same cloud types. Note that during PARCS, measurements were carried out over the coast of Northern Norway, where the influence of local sources of pollution was not negligible (gas flaring from the Melkoya processing facility, presence of shipping activities close to Hammerfest, transport of anthropogenic pollution from Russia) and when a plume containing biomass burning aerosols from huge forest fires in Canada reached

Scandinavia (Chazette et al., 2018). The small values of ice crystals sizes and IWC found in our study may be explained by such polluted situations in comparison to clouds sampled in more pristine conditions in the High Arctic.

Similarly, McFarquhar et al. (2007) investigated the microphysical properties of single-layer stratus clouds over Barrow and Oliktok Point in Alaska as part of the M-PACE (Mixed-Phase

Arctic Cloud Experiment (Verlinde et al., 2007) campaign in fall 2004. They found that the effective radius of ice crystals was 25.2±3.9 µm and nearly independent of the normalized cloud altitude. Korolev and Isaac (2003) investigated mixed-phase clouds associated with frontal systems. They found that the mean volume radius of particles in ice clouds varied between 10 and 17.5 µm. Between -50°C and -30°C where all cloud particles were presumably ice, the

effective radius was found to be about 7 µm which is of similar magnitude to our retrievals despite higher temperatures. They argued that IWC in glaciated clouds decreased with decreasing temperature, from about 100 mg m$^{-3}$ at -5°C to 20 mg m$^{-3}$ at -35 °C. Our observations shown in Fig. 7 do not show any evidence of such behavior for stratiform clouds encountered during PARCS. The vertical profile of IWC cannot be explained solely by the temperature

values but may also be ascribed to the history of the air masses.



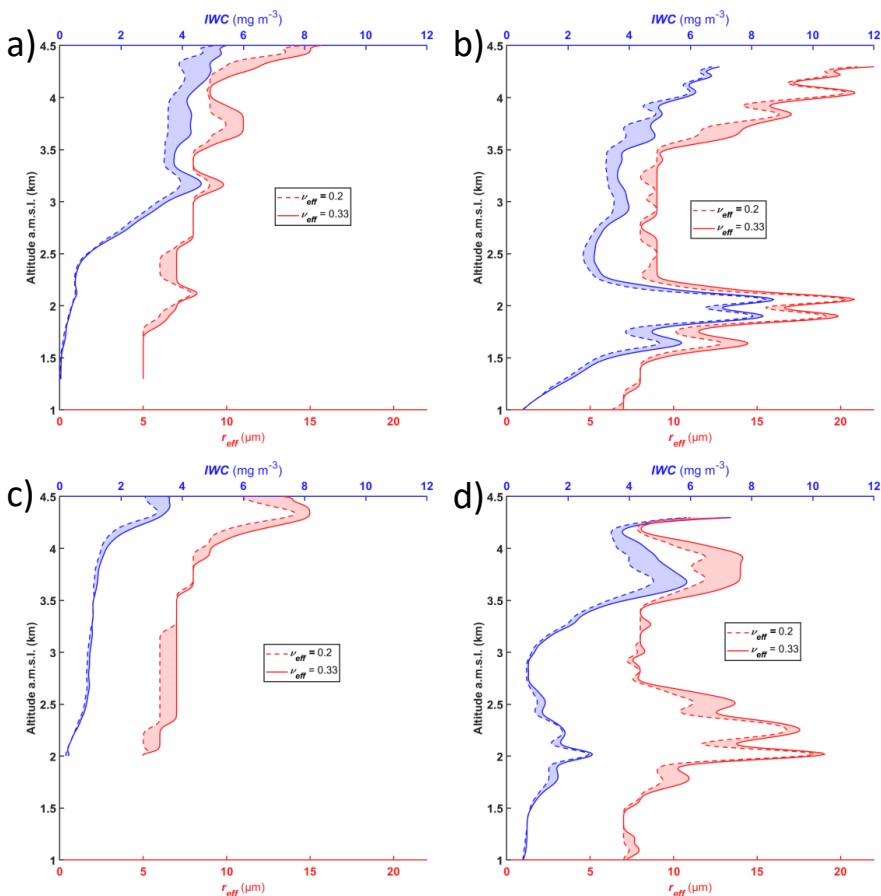

Figure 7. Vertical profiles of linear- and modeling-derived effective radius of ice crystal and ice water content (IWC) on a) 16 May 2016, 17:15 - 18:20 LT, b) 16 May 2016 19:40 - 22:45 LT, c) 17 May 2016 00:00 - 01:15 LT, and d) 17 May 2016 07:30 - 08:20 LT.

## 6 Conclusion

Stratiform clouds were sampled by ground-based lidar in late spring 2016 over the Hammerfest area in northern Norway. In the presence of semi-transparent stratiform clouds with an optical thickness of less than 2.5 at the wavelength of 355 nm, ground-based lidar measurements allow
10    to differentiate the contributions of ice crystals and liquid water pockets embedded in the cloud. The clouds are located just above the atmospheric boundary layer, with a cloud-base between 0.8 and 1.2 km a.m.s.l. where the temperature is below the 0°C isotherm. The inversion of the lidar profiles shows a modest level of depolarization, of the order of 10%, with a negligeable multiple scattering coefficient (<0.95 at the cloud top), suggesting that sampled ice crystals are
15    small and of rather spherical shape. This agrees with Mie computations determining effective



radii between ~5 and 20 µm. The ice water contents are found to be lower than 8 mg m-3. Such small values may be ascribed to more polluted situations compared to pristine conditions in the High Arctic.

The use of vibrational Raman measurements to constrain the elastic lidar equation allows to
remove ambiguities on the restitution of optical properties of semi-transparent ice clouds. It is an alternative approach to airborne in situ measurements limited by the ability to sample the clouds over long periods of time. It is complementary to approaches proposing the coupling of lidar and radar measurements at 95 GHz. Its limitation is mainly on the ability of the lidar to sample the cloud layer along the line of sight and on the assumptions of sphericity of ice
crystals, which have been justified by the observed values of the depolarization ratio. Our approach can nevertheless be easily extended to clouds containing non-spherical ice crystals if we consider the appropriate phase functions, which can be obtained for instance using T-matrix approaches.

**Data availability.** Data from the PARCS Hammerfest campaign can be downloaded from the https://www4.obs-mip.fr/parcs/ database/ database upon request to the first author of the paper.

**Author contributions.** Both authors conceived and participated to the experiment, contributed to the analysis of the lidar data, the conception, and the writing of the manuscript.

**Competing interests.** The authors declare that they have no conflict of interest.

**Acknowledgements.** We thank Xiaoxia Shang, Julien Totems, Yoann Chazette, Nathalie Toussaint, and Sébastien Blanchon for their help during the field experiment. The ULA flights were performed by Franck Toussaint. The Avinor crew of Hammerfest Airport, represented by Hans-Petter Nergård, and the Air Création company are acknowledged for their hospitality. Kathy Law is acknowledged for securing the funding of the Pollution in the ARCtic System
campaign. Computer analyses benefited from access to IDRIS HPC resources (GENCI allocations A011017141 and A013017141) and the IPSL mesoscale computing center.

**Financial support.** This work was supported by the French Institut National de l'Univers (INSU) of the Centre National de la Recherche Scientifique (CNRS) via the French Arctic Initiative and the Commissariat à l'Énergie Atomique et aux Énergies Alternatives (CEA).

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
