# Peer review of "Raman lidar-derived optical and microphysical properties of ice crystals within thin Arctic clouds during PARCS campaign"

_Atmospheric Measurement Techniques, 2023_

## Referee Comment (RC2)

**Review of: "Optical and microphysical properties of ice crystals in Arctic clouds from lidar observations" by Chazette and Raut.**

**General remarks**

The manuscript introduces an approach combining different lidar observed parameters to retrieve ice cloud properties. It makes use of Raman and depolarisation measurements to obtain lidar ratio and thus extinction profiles which are then inverted into ice water content and ice crystal effective radius. The method is applied to a case study of arctic ice clouds and analyses the vertical profiles of the retrieved cloud microphysical properties. As observations in the Arctic, where we can expect a significant amount of pure ice clouds, are still rare, observations and retrievals as presented in the manuscript are of high value. Therefore, the study is well suited for publication in AMT. However, the manuscript lacks in several major issues and, therefore, does not reach its full potential. These major issues have to be reassessed in detail before I a can recommend publishing the manuscript.

First, the manuscript fails in presenting and quantifying the uncertainties of the retrieval. Some issues are named but not quantified. Due to the lack of in situ validations of the retrieved cloud properties, this is a major limitation of the manuscript and does not promote the future application of the method. Second the interpretation of the clouds observed in the case study is partly misleading because the authors misinterpreted the cloud type as far as I can conclude from lidar plot. This leads to an unprecise analysis of the clouds in the discussion.

Below, I compiled a list of comments which have to be considered in a revised version of the paper. There might be some contradictory statements which result from my misinterpretation of the text when first reading the manuscript. I am sure the authors will know how to weight in such cases and how to improve the text to avoid misinterpretations by other readers.

**Major comments**

**Presentation of the retrieval algorithm and uncertainties**
The objective of the AMT journal is to present new methods and elaborate their potential and uncertainties. This was only partly achieved by the authors. First, it is not clear to me if the presented retrieval approach is new or if methods well documented in literature are used? It needs to clearly stated, what are the new aspects of the presented approach and how does it compare to other commonly used cirrus lidar retrieval?

The second objective should be to demonstrate and quantify the uncertainties of the retrieved cloud properties. Only some uncertainty sources are mentioned, but no final quantification is given. In the end, each profile of extinction, IWC and $R_{\text{eff}}$ needs to be presented with an uncertainty range. This is of special importance as there are is no in situ validation available. Here I may summarize some of my thoughts. More details are given in the specific comments below.

- How calibration uncertainties of the lidar lead to uncertainties of the retrieved cloud properties?

- How uncertainties of LR transfer into the retrieved COT and $R_{\text{eff}}$?

- How the assumption of spherical particles affects quantitatively the retrieved COT and $R_{\text{eff}}$?

- How the contamination of measurements by liquid droplets lead to biases in the retrieval? The lidar backscatter plot suggest, that liquid droplets are present.

- How large is the "little" influence of multiple scattering?

**Definition and interpretation of the cloud types**
The use of cloud types which are analyzed in the study and for which the retrieval approach works is not well specified. This addresses two aspects:

a) Starting with the title, it is unclear, what type of Arctic clouds can be analyzed by the presented method. Step by step, the limitations of the retrieval approach reveal. Only ice clouds and only cloud with $\tau < 2$, while in the introduction often stratiform and mixed-phase clouds are discussed. This limitation of the analysis should be made more clear from the beginning, starting with the title.

b) To my point of view, the clouds observed in the case study are not correctly classified, which leads to some misleading interpretation of the data. I only see a few stratiform clouds in the lidar backscatter plot. Most clouds extend through a wide range of altitudes and are rather inhomogeneous. The clouds are also not formed in the boundary layer as indirectly suggested by analyzing the UAV measurements. To me, it rather looks like ice clouds which form in high altitude and which then sediment downward. Some liquid cloud tops (in this case maybe stratiform), where the lidar is immediately extinct, are present in a range of 2 km altitude on 16 May and 2-3km altitude on 17 May. I assume, that the sedimentation of ice crystals from the higher ice clouds leads to a glaciation of most of the lower liquid cloud layers (seeder-feeder effect). Only few mixed-phase clouds can sustain. This makes the cloud situation quite complex. This misclassification has two implication. First, the comparison with cloud properties reported in literature can

become misleading when different cloud types are compared. Second, the retrieval may suffer from the existence of liquid clouds. This impact needs to be discussed.

**List of specific comments**

**P1, title:** The title is partly misleading. It reads very general and to some point suggests to provide a more general statistical evaluation of ice cloud properties. To avoid raising wrong expectation, I suggest to add, that the manuscript a) focuses on demonstrating a method to derive ice clouds properties from lidar and b) that only one case study is analysed.

**P1, L20-22:** Something is missing in this sentence. "large"/"low" optical thickness?

**P1, Abstract:** The abstract does not include what is the major new contribution to the scientific discussion. Is the presented lidar retrieval a "new" technique which is worth to be publish? Or are there major conclusions on the relevance/impact of the cirrus properties? Why should I read this paper?

**P1, L26:** The radiative impact of ice clouds was not calcualted and discussed in the study. I also doubt, that single location lidar observation can provide a "large-scale" estimate of the cloud radiative impact. Additionally I do not understand, what "experimental resources" will be reduced? To what are you comparing? And is this conclusion justified?

**P2, L30:** The 1 km averaging only holds for satellite lidar. Airborne lidar have a much higher spatial resolution.

**P3, L6:** If $\tau < 2$ is the constrain of the study, then this should be highlighted from the beginning including the title.

**P3, L13:** PASCAL was in parallel to ACLOUD in 2017.

**P3, L33:** As the manuscript aims to be a technical paper, at least the major uncertainties documented in the provided literature should be repeated here.

**P4, L3:** Averaging of 1 min is also rather long. Assuming a $10 \, \mathrm{m \, s}-1$ wind speed, you end up to a horizontal resolution of $600 \, \mathrm{m}$. This is similar to what you stated in the motivation for satellite observations and for this you concluded that is is not sufficient to study cloud structures.

**P4, L9:** Should it be rather "cloud-free atmosphere"? I understood, that you have multiple scattering if there are cloud, especially liquid clouds.

**P4, Equation 1:** I recommend to avoid these horizontal brackets in your equations. I first thought this is a normalization. Rather add additional single line equations defining these properties.

**P5, L5:** Can you give a statement, why this assumption (aerosols is negligible) is needed?

**P5, L19:** Does $A_C$ close to 0 mean, you can simplify Eq. 2? If yes, I would show this simplification.

**P6, Equation 6:** Why "A" appears here is without index? If it is $A_C$ and you assume $A_C = 1$, then you end up with Equation 4. Is this intended?

**P7, L2:** This AOT value certainly does not hold in general. Do you refer to Chazette et al. (2018) because it analyses the same measurements/cases? Then please highlight, that the AOT value only holds for this case.

**P7, L9:** In what respect do you use "effective" here? Effective cloud properties merging liquid and ice particles?

**P8, L1:** Does this second method provide identical results? What method did you use in your analysis?

**P8, L6:** This reads strange. I'm not sure, how it is similar to aerosol when there still can be aerosol. As I understand, you substract the molecular DR from the VDR to derive only the contribution of ice crystals. In principle, you end up here with a DR, which characterizes aerosol particles and clouds? Only that you assume, that there is now aerosol particles.

**P8, L11:** add: "Monte Carlo radiative transfer model"

**P8, L24:** The references of the justification using spherical particles is not well chosen. These references describe microphysical schemes in numerical cloud modelling. Assuming spherical shapes in microphysical schemes is something different to assuming spherical particles in radiative transfer models, where the radiative properties matter. This brings me to the question if you can discuss and quantify the uncertainty of your retrieval results with respect to ice crystal shape.

**P9, Equations 15&16:** Why these two parameterisations are given? Which one is applied in your study?

**P9, L18:** Can you show how/where in Eq. 15&16 you derive the link between $v_{\text{eff}}$ and $\mu$?

**P9, L20:** "shape" parameter: Can be misleading, as you discuss on the one hand the shape of the ice crystals and on the other hand the shape of the size distribution. Try to be more precise here to avoid misunderstandings.

**P9, L24:** Why LR is retrieved? I thought, that $R_{\text{eff}}$ will be retrieved from the lidar observations. Similar, it was hard to follow the entire approach and processing chain (e.g., where the Mie calculations are used?). This leads me to suggest adding some kind of overview/flow chart of the entire algorithm to the manuscript. This overview may summarize the most important processing steps, measured parameters and retrieved parameters, and equations/models applied in the retrieval.

**P10, Figure 1:** The sensitivity of LR with $R_{\text{eff}}$ is rather low for large $R_{\text{eff}}$, which are typical for ice clouds. Can you estimate the uncertainty of retrieved $R_{\text{eff}}$ based on the measurement uncertainty of LR?

**P11, L16:** Having negative temperatures does not guaranty the formation of ice clouds. Most likely, super-cooled liquid clouds will from in these conditions. Maybe mixed-phase, if some ice crystals are formed. But still ice formation at temperatures close to $0 \circ$C is not efficient.

As shown in Fig. 3, the clouds do not form in the boundary layer. It rather looks like high ice clouds, which sediment downward. In the altitude range of the UAV, I can not observe any cloud formation. Some liquid cloud tops, where the lidar is immediately extinct, are more in a range of $2 \text{ km}$ altitude on 16 May and 2-3km altitude on 17 May.

**P11, L32:** Your horizontal spatial resolution does not resolve such small scales. What I can identify are stratiform liquid layers. However, the ice cloud is not directly connected to the liquid clouds. I assume, that the sedimentation of ice crystals from higher ice clouds leads to a glaciation of the lower cloud layer in most of the time (seeder-feeder effect). So this is something different to the ice/liquid pockets which are described in the cited literature for stratiform mixed-phase clouds.

**P12, L3-9:** These lines should be moved to somewhere before you discuss the coexistence of liquid and ice. Maybe move to line 28 on page 11.

**P13, Figure 3:** Add day/month/year in figure caption.

**P13, Section 4.2:** I would have liked the synoptic situation discussed before or while introducing your case study. Switch with Section 4.1.? Or merge with a general description of the case in Section 4.1?

**P14, Figure 4:** Indicate where your measurement site is located. Otherwise, the map does not help a lot. As you are investigating clouds, a second panel row with cloud cover, ice water content or humidity might help. This would help to understand how the larger area cloud field looked like. A satellite image might also do it.

**P15, L1:** "write "as indicated in the time series of Fig. 3.""

**P15, L2:** This once more violates your motivation of ground-based lidar measurement having a high spatial resolution. With averaging you end up with horizontal distances far beyond kilometers. Why averaging is needed? How the results would look, if no averaging is applied in order to analyse the detailed horizontal structure/dynamics?

**P15, L3:** In case 2, the lidar reflectivity suggest, that some liquid clouds are present and included in the averaging. How does this affect your retrieved ice cloud properties?

**P15, L14:** What a priori assumption of LR did you assume here? Didn't you had a measured LR?

**P15, L20:** How these uncertainties of LR transfer into the retrieved COT?

**P15, L30:** What about liquid droplets contaminating your retrieval?

**P15, L32:** What size distribution do you retrieve from your method?

**P16, L3:** Can you quantify "little" influence?

**P16, Table 1:** The Klett method was not explained in the methods section. What is different to your approach? What are the known systematic differences?

**P17, Figure 5:** Figure is of poor image quality. I suggest to add a legend for the different lines. I also suggest to check for color-blindness and maybe vary line stile to differentiate the lines.

**P17, Figure 5:** Can you add uncertainty estimates to the extinction profile?

**P18, L12:** This shows only the sensitivity for changing $v_{\text{eff}}$. But how about the uncertainty of retrieved $R_{\text{eff}}$ with respect to uncertainties in LR?

**P18, L15 - P19, L5:** Mioche et al. (2017) characterized low-level mixed-phase clouds. In most cases stratiform mixed-phase clouds in the boundary layer. I don't think, this cloud type can be compared to the ice cloud cases you present here. They are of different nature. This should be emphasized in this comparison.

**P19, L7-10:** You did provide a synoptic analysis of the cloud case. It should be easy to identify the origin of the air mass and potential aerosol particles/pollution sources. Does pollution manifests in your estimates of aerosol particles? I thought, aerosol particle concentration is assumed to be low and neglected in the retrieval.

**P19, L23:** I don't see a stratiform cloud in your case study. These comparisons need to be done with more care.

**P20, L10:** I obviously missed this part in your description of the method. How the scattering of liquid cloud parts is removes/extracted from the ice cloud backscattering in the analysis?

**P20, L15:** This is somehow strange to read. Mie calculations always assume spherical particles. Do I need to understand this conclusion in a way that you would not have had an agreement with Mie calculations, when non-pherical particles are present?

**P21, L1:** Provide uncertainty ranges of your retrieved cloud properties in the conclusion.

**P21, L6:** The lidar retrieval was not validated against in situ observations. I would be hesitant to replace valuable in situ observations by remote sensing observations. In addition, in situ observations do provide much more than only IWC and Reff (e.g., crystal shape, size distribution, roughness,...).

**P21, L7:** It was not shown how the lidar retrieval of the study would complement common lidar/radar retrieval.

**P21, L11-14:** To extend you method to non-spherical ice crystals you would need an assumption on the ice crystals shape. Only knowing that the particles are non-spherical, would not help. What shape/phase function would you assume? How ice crystal size is then translated into IWC

(volume-size ratio,...). This conclusion is very speculative and certainly not "easy".

---

## Author Comment (AC1)

**Responses to Review #1**

**The authors would like to thank the reviewer for his valuable comments which helped improving the quality of the manuscript. Our point-by-point responses to the reviewer's comments appear in bold below.**

*R1.1*

Page 7, line 3 I think you are referring to eq. 6 not 7

**The sentence has been fixed.**

*R1.2*

Page 9, line 16 to 18: if μi = 0 why is veff =1/3 (and not 1)?

**There was an error in the equation on line 18. It has been corrected:**

$$\nu_{eff} = \frac{1}{\mu_i + 3} = \frac{1}{3}$$

*R1.3*

Page 9, calculation of LR via Mie code: please state that this is only a (rough?) approximation as the ice crystals are not spherical.

**This is indeed a rough approximation for ice crystals, which are generally far from spherical. Thanks to lidar observations, this non-sphericity is characterized by the depolarization rate, which is generally over 30%, often 50%. In our case, we have values of the order of 10%. This suggests that the crystals are approaching a spherical shape. This is the assumption we have made. We have supplemented the text where we introduce the use of a Mie code:**

"In order to assess the vertical profiles of ice crystal effective radius ($r_{eff}$) and ice water content (IWC), we use a Mie code assuming... **Ice crystals are generally not spherical. Nevertheless, for the clouds sampled in this study, the ICDR is ~10% (Sect. 4.3), far from the values associated with highly non-spherical crystals whose ICDR is between 30 and 50%...**"

*R1.4*

Page 11, l16 "ice cloud formation highly probable" Please consider rewording. To my knowledge, in INP sparse regions supercooled liquid clouds dominate even at much lower temperatures. However, a recent paper that describes ice formation at temperatures slightly below 0C might be this

https://www.pnas.org/doi/pdf/10.1073/pnas.2021387118

**Thank you for your comment. We have added the reference and clarified our comments accordingly:**

"With the advection of moist air masses over the site, the formation of ice clouds is therefore highly probable above 0.8 km a.m.s.l. This value is within the range of the bottom height of boundary layer mixed phase clouds in the western Arctic (McFarquhar et al., 2007; Maillard et al., 2021). **At temperatures close to those encountered here, i.e. slightly below 0°C, Luke et al. (2021) have recently shown that secondary ice formation can significantly increase the amount of ice crystals in clouds in the Arctic.**"

*R1.5*

Fig 3: I agree that we see a highly complex pattern. Hence, is this really a stratus cloud? The cloud at about 2km altitude from UT 18 to 23 probably yes.

**This is a good remark also made by reviewer 2. We have made corrections where necessary. There are probably Ci composed of ice crystals that precipitate. This could lead to glaciation of the lower liquid layers, as suggested by reviewer 2.**

*R1.6*

Fig 3: Your VDR has a tendency to show high values above 6km, regardless whether a cloud is obvious in ABR or not. This may be a thin / subvisual cirrus. But what about a typical insecurity of VDR and ABR in about 6km altitude for the night 16 to 17 May?

**Indeed, there are certainly thin cirrus clouds above 6 km and this is in line with comment R1.5. ABR is a fairly raw data, associated with a relative error of less than 5% at the altitudes considered for the lidar used and the optical thicknesses encountered. VDR is a ratio of channels, so the relative error is higher. It is only used as an indicator. The absolute error is around 1% (Chazette et al., 2012). It is highly dependent on the optical thickness of the atmosphere. We have added an error calculation on the retrieved optical properties.**

*R1.7*

Page 11, line 25: variability … due to wind shear. In the boundary layer I understand this point. What about the free troposphere? Maybe you could describe your measurement site a bit. E.g. are you surrounded by mountains? What was the synoptic situation / main wind direction etc.

… Ah, I see that you describe this in section 4.2. Still you may add a short description of your site and show this prior to the lidar results?

**It was indeed described in sub-section 4.2. We have followed the reviewer's advice and moved this subsection back to the beginning of Section 4. The wind directions are already given in Fig. 2. We have added a description of the site at the beginning of Subsection 4.2 (now 4.1):**

**"The lidar measurements were obtained near Hammerfest airport on the island of Kvaløya, which lies in a south-west/north-east trough at an altitude of ~90 m a.m.s.l. The site is bordered by relief peaking at around 360 m a.m.s.l. in the north-west and reaching up to 1045 m a.m.s.l. in the south-east. These reliefs can therefore significantly influence flows over the site by generating wind shears.**

*R1.8*

Page 15 line 30.: relatively low depolarization values for Arctic cirrus have recently been found by Nakoudi. They speculate on a latitude dependence of depolarization. However, as your clouds are low and warm this may not be 1:1 comparable. Still I am not too surprised on your findings.

https://www.mdpi.com/2072-4292/13/22/4555

**We have added this reference, which compares well with our results.**

**R1.9**

Table 1: eq 13 is the "backscatter weighted LR". Eq 12 is a constant LR. If the LR according to eq. 13 is larger than for eq 12 this means (to me, maybe I am wrong) that the thicker parts of the clouds (high backscatter) have a higher LR and a large concentration (high beta) of smaller particles (your Fig 1.) I am wondering whether this does make sense. What do you think?

**Yes, if we look only at equation 13. In fact, when we hang a constant LR, constrained by optical thickness, we should find the same thing. Equation 13 uses the N2 Raman channel, which doesn't reach as far as the elastic scattering channel (altitude ranges in brackets in the table). The fact that equation 13 gives a higher LR is therefore linked to a non-negligible contribution from the top of the cloud, which may be associated with lower LRs. However, we must be cautious with the uncertainties on LRs.**

**R1.10**

Fig 5: can you please state briefly how this has been calculated? This is the solution of the Raman channel for alpha, I assume. And beta was taken from Klett? The lidar rations in Fig 5 are larger than in Table 1. Why?

**The LRs in Fig. 5 are obtained by calculating the backscattering coefficient via the coupling between the N2 Raman channel and the elastic channel. The extinction coefficient is derived from the N2 Raman channel. The LR in the table is an average value of the LR profile weighted by the extinction profile (Equation 13, now equation 10). High extinction coefficients therefore mostly drive the average LR. This explains the apparent differences noted by the reviewer.**

**R1.11**

Page 19, lines 6-10: While everything (local pollution, Norwegian gas flaring, pollution from Russia and Canadian forest fires can occur), they will probably not manifest in the same night of observations. I would skip the speculation on the forest fires or Russian pollution unless you have a clear hint that you have seen it in the lidar data. Instead, you see an ice cloud at warm conditions. Maybe the growth rate is simply slower?

**Just before the cloud period, we noted a significant change in the nature of aerosols in the atmospheric column, with fire aerosols aloft. There are therefore several aerosol natures**

passing over the site, as shown in the reference quoted in the article (Chazette et al., 2018). Clouds encompass these different layers.

---

## Author Comment (AC2)

**Responses to Review #2**

**The authors would like to thank the reviewer for his valuable comments which helped improving the quality of the manuscript. Our point-by-point responses to the reviewer's comments appear in bold below.**

**Major comments**

**Presentation of the retrieval algorithm and uncertainties**

*R2.1*

The objective of the AMT journal is to present new methods and elaborate their potential and uncertainties. This was only partly achieved by the authors. First, it is not clear to me if the presented retrieval approach is new or if methods well documented in literature are used? It needs to clearly stated, what are the new aspects of the presented approach and how does it compare to other commonly used cirrus lidar retrieval?

**To our knowledge, the Raman lidar approach has not been used for ice clouds. The advantage of a Raman lidar is that the LR profile can be traced over a large part of the cloud, as extinction and backscatter are determined independently. We've clarified the title accordingly by adding "Raman" to the title and by taking into account that our method requires thin clouds:**

**"Raman lidar-derived optical and microphysical properties of ice crystals within thin Arctic clouds during PARCS campaign"**

The second objective should be to demonstrate and quantify the uncertainties of the retrieved cloud properties. Only some uncertainty sources are mentioned, but no final quantification is given. In the end, each profile of extinction, IWC and $R_{eff}$ needs to be presented with an uncertainty range. This is of special importance as there are is no in situ validation available. Here I may summarize some of my thoughts. More details are given in the specific comments below.

*R2.2*

- How calibration uncertainties of the lidar lead to uncertainties of the retrieved cloud properties?

    **A Monte Carlo error propagation model was used to calculate uncertainties in COT and LR. This has been added in the revised manuscript.**

*R2.3*

- How uncertainties of LR transfer into the retrieved COT and $R_{eff}$?

**LR has no effect on COT with the method used since the extinction and backscatter coefficients are determined independently. It does, however, have an effect on Reff, and the error has been evaluated using the Monte Carlo results and Fig. 1.**

*R2.4*

- How the assumption of spherical particles affects quantitatively the retrieved COT and $R_{eff}$?

  **For the COT, there is no effect: it is indeed directly derived from the optical measurements without any assumption on the crystals' shape**

  **There are no direct measurements of crystal shapes, and assuming one particular crystal shape could lead to significant error. The same issues also arise with satellite inversion techniques. The assumption of crystal sphericity comes from our measurements of low depolarization.**

*R2.5*

- How the contamination of measurements by liquid droplets lead to biases in the retrieval? The lidar backscatter plot suggest, that liquid droplets are present.

  **Yes, there are pockets of liquid water as mentioned in the text, but as explained, we get rid of them thanks to the VDR.**

*R2.6*

- How large is the "little" influence of multiple scattering?

  **Multiple scattering mainly influences the top of the cloud, as it lengthens the optical path. Moreover, we have calculated a multiple scattering coefficient close to 1 (>0.95). There is therefore a very small influence of multiple scattering in this case.**

  **We have corrected this by replacing "little" by "negligible".**

**Definition and interpretation of the cloud types**

The use of cloud types which are analyzed in the study and for which the retrieval approach works is not well specified. This addresses two aspects:

*R2.7a*

Starting with the title, it is unclear, what type of Arctic clouds can be analyzed by the pre-sented method. Step by step, the limitations of the retrieval approach reveal. Only ice clouds and only cloud with $\tau < 2$, while in the introduction often stratiform and mixed-phase clouds are discussed.

This limitation of the analysis should be made more clear from the beginning, starting with the title.

**We have added "thin" in the title and "ice cloud" is already in the title. Furthermore, the abstract makes it clear that these are semi-transparent clouds. The introduction clearly states that the clouds are made of ice and are semi-transparent. On the other hand, in response to the reviewer's helpful comment, we have removed the reference to stratiform clouds for the PARCS campaign throughout the article.**

*R2.7b*

To my point of view, the clouds observed in the case study are not correctly classified, whichleads to some misleading interpretation of the data. I only see a few stratiform clouds in the lidar backscatter plot. Most clouds extend through a wide range of altitudes and are rather inhomogeneous.

**We thank the reviewer for this comment. We agree and have removed references to "stratiform clouds", see previous comment.**

*R2.7c*

The clouds are also not formed in the boundary layer as indirectly suggested by analyzing the UAV measurements. To me, it rather looks like ice clouds which form in high altitude and which then sediment downward. Some liquid cloud tops (in this case maybe stratiform), where the lidar is immediately extinct, are present in a range of 2km altitude on 16 May and 2-3km altitude on 17 May. I assume, that the sedimentation of ice crystals from the higher ice clouds leads to a glaciation of most of the lower liquid cloud layers (seeder-feeder effect). Only few mixed-phase clouds can sustain. This makes the cloud situation quite complex. This misclassification has two implication. First, the comparison with cloud properties reported in literature can become misleading when different cloud types are compared. Second, the retrieval may suffer from the existence of liquid clouds. This impact needs to be discussed.

**The word "formation" was not appropriate, so we have replaced it with "presence". The description given by the reviewer is relevant to be included in the article. It has been added in subsection 4.1:**

**"Some stratiform clouds may occur at 2 km altitude on 16 May and in the range 2-3 km altitude on 17 May and can be detected by a strong attenuation of the lidar signal at their tops. This might indicate the presence of supercooled liquid droplets at the top of mixed-phase clouds, as often reported in the Arctic region (Mioche et al., 2017; Mc Farquhar et al., 2017). Higher clouds (2-6 km altitude) are also detected by the lidar. The sedimentation**

**of ice crystals from those higher ice clouds leads to a glaciation of most of the lower liquid cloud layers (seeder-feeder effect, Fernandez-Gonzalez et al., 2015)."**

**The type of clouds observed is complex difficult to compare with what is described in the literature. We have therefore compared them with what currently exists. The presence of liquid water clouds has no impact on the results, as we have excluded pockets of liquid water using VDR values.**

**List of specific comments**

**R2.8**

P1, title: The title is partly misleading. It reads very general and to some point suggests to provide a more general statistical evaluation of ice cloud properties. To avoid raising wrong expectation, I suggest to add, that the manuscript a) focuses on demonstrating a method to derive ice clouds properties from lidar and b) that only one case study is analysed.

**The title has been changed to reflect the reviewer's comments: "Raman lidar-derived optical and microphysical properties of ice crystals within thin Arctic clouds during PARCS campaign"**

**R2.9**

P1, L20-22: Something is missing in this sentence. "large"/"low" optical thickness?

**The correction has been done.**

**R2.10**

Abstract: The abstract does not include what is the major new contribution to the scientific discussion. Is the presented lidar retrieval a "new" technique which is worth to be publish? Or are there major conclusions on the relevance/impact of the cirrus properties? Why should I read this paper

**The abstract has been rewritten following the suggestions of the reviewer (see below).**

**R2.11**

P1, L26: The radiative impact of ice clouds was not calcualted and discussed in the study. I also doubt, that single location lidar observation can provide a "large-scale" estimate of the cloud radiative impact. Additionally I do not understand, what "experimental resources" will be reduced? To what are you comparing? And is this conclusion justified?

**The reviewer is right, we have therefore removed the last sentence and corrected the abstract to take his/her other comments into account:**

**"Cloud observations in the Arctic are still rare, which requires innovative observation technics to assess ice crystal properties. We present an original approach using the Raman lidar measurements applied to a case study in northern Scandinavia. The vertical profiles of the optical properties, effective radius of ice crystals and ice water content (IWC) in Arctic semi-transparent clouds were assessed using quantitative ground-based Raman lidar measurements at 355 nm performed from 13 to 26 May 2016 in Hammerfest (north of Norway, 70° 39′ 48″ North, 23° 41′ 00″ East). The field campaign was part of the Pollution in the ARCtic System (PARCS) project of the French Arctic Initiative. The presence of semi-transparent clouds was noted on 16 and 17 May. The cloud base was located just above the atmospheric boundary layer where the 0°C isotherm reached around 800 m above the mean sea level (a.m.s.l.). To ensure the best penetration of the laser beam into the cloud, we selected case studies with cloud optical thickness (COT) lower than 2 and out of supercooled liquid pockets. Lidar-derived multiple scattering coefficients were found to be close to 1 and ice crystal depolarization around 10%, suggesting that ice crystals were small and had a rather spherical shape. Using Mie computations, we determine effective radii between ~5 and 20 μm in the clouds for ice water contents between 1 and 8 mg m$^{-3}$, respectively. The uncertainties on the effective radius and ice water content are in average of 2 μm and 0.65 mg m$^{-3}$, respectively."**

*R2.12*

P2, L30: The 1km averaging only holds for satellite lidar. Airborne lidar have a much higher spatial resolution.

**The temporal resolutions are similar, but to invert the data, profiles have to be averaged. We agree that better resolution can be achieved with airborne lidar. We have made some changes in this direction.**

*R2.13*

P3, L6: If $\tau$ <2 is the constrain of the study, then this should be highlighted from the beginning including the title.

**We added "thin clouds" in the title and specified the COT limit value in the abstract.**

*R2.14*

P3, L13: PASCAL was in parallel to ACLOUD in 2017.

**This has been corrected.**

*R2.15*

P3, L33: As the manuscript aims to be a technical paper, at least the major uncertainties documented in the provided literature should be repeated here.

**Following the major comment of the reviewer, we have recalculated the error budget when presenting the results and then removed the sentence.**

*R2.16*

P4, L3: Averaging of 1min is also rather long. Assuming a 10ms−1 wind speed, you end up to a horizontal resolution of 600m. This is similar to what you stated in the motivation for satellite observations and for this you concluded that is is not sufficient to study cloud structures.

**The two measurement platforms are subject to the same wind effects. In addition, the displacement of a satellite is of the order of 7 kms-1 and processing the information requires the averaging of several profiles. Resolution is highly dependent on the product being measured.**

*R2.17*

P4, L9: Should it be rather "cloud-free atmosphere"? I understood, that you have multiple scattering if there are cloud, especially liquid clouds.

**The choice of telescope aperture limits multiple scattering, whatever the composition of the atmosphere.**

*R2.18*

P4, Equation 1: I recommend to avoid these horizontal brackets in your equations. I first thought this is a normalization. Rather add additional single line equations defining these properties.

**The corrections have been done.**

*R2.19*

P5, L5: Can you give a statement, why this assumption (aerosols is negligible) is needed?

**Without this assumption, COT would have a particulate component and previous measurements show that aerosol contribution is negligible.**

*R2.20*

P5, L19: Does $A_C$ close to 0 mean, you can simplify Eq. 2? If yes, I would show this simplification.

**The equations are given in a general form and then simplified according to assumptions. In clouds, the spectral dependence of extinction is close to 0.**

*R2.21*

P6, Equation 6: Why "A" appears here is without index? If it is $A_C$ and you assume $A_C = 1$, then you end up with Equation 4. Is this intended?

**We used the previous assumption (Ac = 0) and made the correction. We simplified the system of equations for the sake of clarify. We've kept only those equations that are actually useful for tracing the microphysical properties of clouds.**

*R2.22*

P7, L2: This AOT value certainly does not hold in general. Do you refer to Chazette et al. (2018) because it analyses the same measurements/cases? Then please highlight, that the AOT value only holds for this case.

**Raman lidar can be used to calculate the AOT below the cloud layer and it corresponds to the cloud-free situation. We have clarified.**

*R2.23*

P7, L9: In what respect do you use "effective" here? Effective cloud properties merging liquid and ice particles?

**"Effective" is defined as the optical properties including multiple scattering (see equation 9), as a common practice in the scientific literature. It does not assume mixing between liquid droplets and ice crystals.**

*R2.24*

P8, L1: Does this second method provide identical results? What method did you use in your analysis?

**Yes, it should lead to the same result. For the sake of clarity, we have removed Section 2.3.3 that is not useful for the main outcome of the paper.**

*R2.25*

P8, L6: This reads strange. I'm not sure, how it is similar to aerosol when there still can be aerosol. As I understand, you substract the molecular DR from the VDR to derive only the contribution of ice crystals. In principle, you end up here with a DR, which characterizes aerosol particles and clouds? Only that you assume, that there is now aerosol particles.

**It has been assumed that there are no aerosols in the cloud, as their backscatter coefficient is negligible compared to the one of ice crystals. Equation (14) ((11) in the revised manuscript) does not necessarily mean that there are aerosols. The expression is just the same as for aerosols.**

**R2.26**

P8, L11: add: "Monte Carlo radiative transfer model"

**The correction has been done.**

**R2.27**

P8, L24: The references of the justification using spherical particles is not well chosen. These references describe microphysical schemes in numerical cloud modelling. Assuming spherical shapes in microphysical schemes is something different to assuming spherical particles in radiative transfer models, where the radiative properties matter. This brings me to the question if you can discuss and quantify the uncertainty of your retrieval results with respect to ice crystal shape.

**The justification for the spherical hypothesis comes from the depolarization level in section 4.3, which is the only element available for remote sensing.**

**R2.28**

P9, Equations 15&16: Why these two parameterisations are given? Which one is applied in your study?

**These are two ways of expressing the same distribution, the second being normalized by the total number of crystals. The link between the two equations is explained in the text.**

**R2.29**

P9, L18: Can you show how/where in Eq. 15&16 you derive the link between $\nu_{eff}$ and $\mu$?

**There was an error in the equation. It has been corrected.**

**R2.30**

P9, L20: "shape" parameter: Can be misleading, as you discuss on the one hand the shape of the ice crystals and on the other hand the shape of the size distribution. Try to be more precise here to avoid misunderstandings.

**This is the name given by the community to this parameter of a Gamma distribution. It's better to keep it. We used the term "shape parameter" to avoid any confusion.**

**R2.31**

P9, L24: Why LR is retrieved? I thought, that $R_{eff}$ will be retrieved from the lidar observations. Similar, it was hard to follow the entire approach and processing chain (e.g., where the Mie calculations are used?). This leads me to suggest adding some kind of overview/flow chart of the

entire algorithm to the manuscript. This overview may summarize the most important processing steps, measured parameters and retrieved parameters, and equations/models applied in the retrieval.

**As explained in the text, the LR calculated from the Mie code outputs for different effective radii is compared with the LR measured at each altitude to estimate the cloud effective radius and ice water content. As suggested by the reviewer, we have added a flow chart.**

[Figure]

P10, Figure 1: The sensitivity of LR with $R_{eff}$ is rather low for large $R_{eff}$, which are typical for ice clouds. Can you estimate the uncertainty of retrieved $R_{eff}$ based on the measurement uncertainty of LR?

**We estimated the uncertainties using a Monte Carlo error propagation procedure.**

P11, L16: Having negative temperatures does not guaranty the formation of ice clouds. Most likely, super-cooled liquid clouds will from in these conditions. Maybe mixed-phase, if some ice crystals are formed. But still ice formation at temperatures close to 0◦C is not efficient.

**We agree with the reviewer. The term "formation" was inappropriate. Our sentence was misleading, and we have corrected it.**

As shown in Fig. 3, the clouds do not form in the boundary layer. It rather looks like high ice clouds, which sediment downward. In the altitude range of the UAV, I can not observe any cloud

formation. Some liquid cloud tops, where the lidar is immediately extinct, are more in a range of 2km altitude on 16 May and 2-3km altitude on 17 May.

**See previous response.**

*R2.35*

P11, L32: Your horizontal spatial resolution does not resolve such small scales. What I can identify are stratiform liquid layers. However, the ice cloud is not directly connected to the liquid clouds. I assume, that the sedimentation of ice crystals from higher ice clouds leads to a glaciation of the lower cloud layer in most of the time (seeder-feeder effect). So this is something different to the ice/liquid pockets which are described in the cited literature for stratiform mixed-phase clouds.

**Our vertical resolution is 15 m in agreement with the size of the liquid pockets.**

*R2.36*

P12, L3-9: These lines should be moved to somewhere before you discuss the coexistence of liquid and ice. Maybe move to line 28 on page 11.

**Thisparagraph uses the apparent backscatter ratio and the VDR, so we think it's best to put it after the presentation of this data.**

*R2.37*

P13, Figure 3: Add day/month/year in figure caption.

**The correction has been done.**

*R2.38*

P13, Section 4.2: I would have liked the synoptic situation discussed before or while introducing your case study. Switch with Section 4.1.? Or merge with a general description of the case in Section 4.1?

**The reviewer is right, we have switched the two subsections.**

*R2.39*

P14, Figure 4: Indicate where your measurement site is located. Otherwise, the map does not help a lot. As you are investigating clouds, a second panel row with cloud cover, ice water content or humidity might help. This would help to understand how the larger area cloud field looked like. A satellite image might also do it.

**Relative humidity remains very high throughout the period, as shown in the figure below, but this does not provide any information on cloud type. Both the relative humidity and**

**cloud structures are large-scale disturbances. The reviewer is right that it is preferable to indicate the location of the site, which is what has been done on the new figure (Fig. 2 now).**

[Figure]

*R2.40*

P15, L1: "write "as indicated in the time series of Fig. 3.""
**It has been added.**

*R2.41*

P15, L2: This once more violates your motivation of ground-based lidar measurement having a high spatial resolution. With averaging you end up with horizontal distances far beyond kilometers. Why averaging is needed? How the results would look, if no averaging is applied in order to analyse the detailed horizontal structure/dynamics?
**Structures can be described with high temporal resolution, but this requires the profiles to be averaged to derive the effective radius reff. The final resolution depends on the parameter of interest. We can't go back to a figure similar to Fig. 3 (now Fig. 4) for reff or IWC.**

*R2.42*

P15, L3: In case 2, the lidar reflectivity suggest, that some liquid clouds are present and included in the averaging. How does this affect your retrieved ice cloud properties?

**Liquid pockets have been excluded from the averaging using VDR. In presence of liquid droplets, the VDR is very low.**

*R2.43*

P15, L14: What a priori assumption of LR did you assume here? Didn't you had a measured LR?

**We have removed the column in Table 1.**

*R2.44*

P15, L20: How these uncertainties of LR transfer into the retrieved COT?

**There is no effect on Raman-derived COT.**

*R2.45*

P15, L30: What about liquid droplets contaminating your retrieval?

**Using the ICDR values assessed here, the presence of ice turns out to be certain. There may be liquid water mixed in with the ice, but it is impossible to conclude with remote sensing measurements. As explained, pockets of liquid water are well identified and not taken into account, and liquid water are rather found at the top of clouds and not distributed over their whole thickness.**

*R2.46*

P15, L32: What size distribution do you retrieve from your method?

**We use the method described earlier (Section 3), which is now represented by a flow chart. The available reffs (from a Gamma distribution) depend on the LR value and the uncertainty at which it is obtained. This becomes clearer with the added Monte Carlo study.**

*R2.47*

P16, L3: Can you quantify "little" influence?

**"Little" has been replaced by "negligible".**

*R2.48*

P16, Table 1: The Klett method was not explained in the methods section. What is different to your approach? What are the known systematic differences?

**The Klett method is well-referenced and requires knowledge of LR. The use of a Raman lidar does not need this constraint. We only use it to confirm our results.**

*R2.49*

P17, Figure 5: Figure is of poor image quality. I suggest to add a legend for the different lines.
**A Legend has been added.**

*R2.50*

I also suggest to check for color-blindness and maybe vary line stile to differentiate the lines.
**We have replotted the different curves taken into account this comment.**

*R2.51*

P17, Figure 5: Can you add uncertainty estimates to the extinction profile?
**Uncertainties have been added with filled areas around averaged profiles.**

*R2.52*

P18, L12: This shows only the sensitivity for changing $v_{eff}$. But how about the uncertainty of retrieved $R_{eff}$ with respect to uncertainties in LR?
**The uncertainty calculation was added using a Monte Carlo approach named called "statistical error propagation" to avoid confusion with Monte Carlo calculations in radiative transfer. We've added a figure showing error profiles as a function of altitude where ices crystals are located.**

*R2.53*

P18, L15 - P19, L5: Mioche et al. (2017) characterized low-level mixed-phase clouds. In most cases stratiform mixed-phase clouds in the boundary layer. I don't think, this cloud type can be compared to the ice cloud cases you present here. They are of different nature. This should be emphasized in this comparison.
**We have added this caution: "They have analyzed the vertical distribution of microphysical properties, in most cases stratiform mixed-phase clouds in the boundary layer, using in situ measurements from four airborne spring campaigns in the European Arctic between 2004 and 2010. For these clouds of different nature in the south-western regime…"**

*R2.54*

P19, L7-10: You did provide a synoptic analysis of the cloud case. It should be easy to identify the origin of the air mass and potential aerosol particles/pollution sources. Does pollution manifests in your estimates of aerosol particles? I thought, aerosol particle concentration is assumed to be low and neglected in the retrieval.

**As shown in Chazette et al. (2018) there are different types of aerosols present above the site, but mainly in the boundary layer, so below the clouds. Except the transport of biomass burning aerosols, the amount of aerosols is negligible in the free troposphere, even more when compared to the extinguishing properties of clouds.**

*R2.55*
P19, L23: I don't see a stratiform cloud in your case study. These comparisons need to be done with more care.
**We have withdrawn the mention "stratiform".**

*R2.56*
P20, L10: I obviously missed this part in your description of the method. How the scattering of liquid cloud parts is removes/extracted from the ice cloud backscattering in the analysis?
**We positioned outside the liquid water pockets using the VDR.**

*R2.57*
P20, L15: This is somehow strange to read. Mie calculations always assume spherical particles. Do I need to understand this conclusion in a way that you would not have had an agreement with Mie calculations, when non-pherical particles are present?
**As explained, since depolarization is low, we have assumed that the ice crystals are spherical. We could do T-Matrix calculations, but we'd also have to assume a crystal shape. We agree that this is a strong assumption.**

*R2.58*
P21, L1: Provide uncertainty ranges of your retrieved cloud properties in the conclusion.
**We have added uncertainties.**

*R2.59*
P21, L6: The lidar retrieval was not validated against in situ observations. I would be hesitant to replace valuable in situ observations by remote sensing observations. In addition, in situ observations do provide much more than only IWC and Reff (e.g., crystal shape, size distribution, roughness,...).
**The reviewer is right. We have removed the reference to "in situ".**

*R2.60*

P21, L7: It was not shown how the lidar retrieval of the study would complement common lidar/radar retrieval.

**As mentioned, this is an alternative method that allows direct determination of the LR and uses the VDR to identify the presence of ice crystals. Its advantage lies in the use of a single instrument, but its limitation lies in the lidar's ability to penetrate the cloud layer. This is closely linked to the type of instrument used. In the presence of semi-transparent clouds, this approach can be used instead of the more classical lidar-radar retrieval.**

*R2.61*

P21, L11-14: To extend you method to non-spherical ice crystals you would need an assumption on the ice crystals shape. Only knowing that the particles are non-spherical, would not help. What shape/phase function would you assume? How ice crystal size is then translated into IWC (volume-size ratio,...). This conclusion is very speculative and certainly not "easy".

**The reviewer is right. You do need to assume a shape function before performing the calculations. We have added this point. An additional difficulty is that the symmetry of crystals does not necessarily lead to a two-dimensional phase function. This is a very difficult case to deal with, since we also need to know their orientation in space.**

---

## Author Response (AR2)

**Responses to editor**

Sorry, maybe I overlooked it in the first version, but in eq 1 and 2 you are using different names for /beta_app : (range-corrected) lidar signal in eq2, which I like more, and "total apparent backscatter coefficient" in eq 1 (which I think is confusing, it should be the range-corrected lidar signal). Can you please check at the end the equations 1 to 12 for consistency? Maybe introduce the 1/z^2 in eq 1 and 2 and keep a consistent wording.

**It is not exactly the range-corrected lidar signal, as we also correct for atmospheric transmission. That is why we call it "apparent". The solid angle is also corrected (range-corrected lidar signal in $1/z^2$). We have specified this point in p4l28:**

**"After the molecular transmission*, background radiance and solid angle are* corrected, …"**

Can you please (once again? – sorry maybe my fault) explain, why the apparent backscatter ratio drops below 1, such that the reader understands the values in fig 3.a quicker.

The ABR drops below 1 because the apparent backscatter coefficient (equation 1) is attenuated by the atmospheric transmission due to aerosols and clouds. We have added this point in p14l24:

**"The apparent backscatter ratio ($\beta_{app}/(C \cdot \beta_m)$*), including the attenuation due to aerosols and clouds*, …"**

The flow chart as response to reviewer No 2 is helpful. I support showing this in the final version. Yes,
**Yes, it's a good idea and we have kept it in the final version.**

Otherwise: good work!

**Thank you.**